

# GDPRValidator: a tool to enable companies using cloud services to be GDPR compliant

M. Emilia Cambronero[1], Miguel A. Martínez[1], José Luis de la Vara[1], David Cebrián[2] and Valentín Valero[1]

[1] Computer Science Department, Universidad de Castilla La Mancha, Albacete, Spain
[2] Santander Private Banking International, Madrid, Spain

## ABSTRACT

This article presents a tool called GDPRValidator that aims to assist small and medium-sized enterprises (SMEs) that have migrated their services, or a part of them, to the cloud to be General Data Protection Regulation (GDPR) compliant when they manage and store employees' or customers' data in the cloud. As these companies have a limited budget to hire legal experts to guide them in complying with GDPR, the main objective of this tool is to help SMEs to be more competitive by saving a considerable amount of money. By using GDPRValidator, these companies can learn and begin the GDPR compliance process by themselves and decide whether it will be necessary to hire GDPR legal experts in the end. GDPRValidator implements a process that aids companies in compliance analysis and validation and generates a series of documents with recommendations. These documents do not guarantee full GDPR compliance, but they can help the company better understand the regulation and improve its data management strategies. In order to validate the efficiency and efficacy of the tool, two SMEs have used it and provided feedback about its perceived ease of use and its perceived usefulness for understanding and complying with GDPR. The results of the validation showed that, for both companies, the degree of perceived usefulness and ease of use of GDPRValidator is quite good. All the scores expressed agreement.

## INTRODUCTION

Today, it is truer than ever to state that technology cannot achieve its full potential without user trust, and establishing strong data protection measures is key to building trust. Furthermore, due to the development of technologies based on distributed storage and processing, such as cloud computing, it has become more difficult to control, monitor, and safeguard the personal data handled by these types of infrastructure, making data breaches more likely. For this technology to achieve its full potential and deliver maximum benefit, it must provide a transparent and auditable environment. Data protection is therefore essential in achieving this goal. Migration of IT workloads to the cloud is

Corresponding author
M. Emilia Cambronero,
memilia.cambronero@uclm.es

associated with security risks and responsibilities. For this reason, organizations are establishing security and compliance governance procedures to manage their security duties in the cloud.

As of 2022, over 60% of all corporate data will be stored in the cloud, according to Statista (*Statista, 2022*). It reached 30% in 2015 and has continued to rise as companies move their resources to the cloud. Therefore, the aim is to improve security and reliability while advancing business agility. Thus, the main success factor for organizations in their cloud operations is to balance cloud benefits and security tasks. Cloud security requires a combination of legal, technical, operational, and governance measures, and effective cloud security governance requires policies, procedures, and processes. In this context, the European Union (EU) adopted the General Data Protection Regulation (GDPR) (*GDPR, 2016*) in 2016. GDPR replaces a Data Protection Directive from 1995, which was adopted when the Internet was relatively young. European Union countries must adapt their national laws to enforce the GDPR, but there have been many problems in this process, for instance, the lack of staff trained in GDPR (*GDPR, 2020b*). In June 2020, the European Commission published its first report on the evaluation and review of the GDPR (*GDPR, 2020a*). This report comes two years after the regulation became applicable on 25 May 2018. As stipulated by GDPR article 97 (*GDPR.EU, 2018*), it particularly assesses the following issues: (1) the application and functioning of the rules on the transfer of personal data to third countries and international organizations; (2) the application and functioning of the rules on cooperation and consistency; and finally, (3) the issues raised over the past two years by various actors.

Articles 24 and 25 of the GDPR introduce the figure of the data controller. This controller is responsible for implementing technical and organizational measures to ensure and demonstrate that data processing complies with the regulation, and these measures can be reviewed and updated when necessary. Furthermore, GDPR articles 37, 38, and 39 define the role and position of the Data Protection Officer (DPO) (*GDPR.EU, 2022c*). The DPO is responsible for overseeing an organization's data protection strategy and implementation. However, a DPO only needs to be appointed when certain conditions are met, according to GDPR article 37. Generally, these conditions arise when a company's activity requires large-scale data processing or when the processed data is sensitive or belongs to special categories, according to articles 9 and 10 of GDPR. Nevertheless, it is recommended that small and medium enterprises (SMEs) that plan to enter international markets hire a DPO (*ComplianceJunction, 2018*) to assist in the expansion process. The expense of hiring a full-time DPO (*GDPR.EU, 2022c*) may not be feasible for SMEs, but the DPO can be hired or shared by several SMEs in such cases.

In this article, we propose the `GDPRValidator` tool to help SMEs in assessing their GDPR compliance. Its purpose is to make compliance with GDPR easier for data controllers and to simplify the work of DPOs. Thus, the main motivation is to assist them to be more competitive since by using `GDPRValidator` these companies can learn and begin the GDPR compliance process by themselves, better understand the regulation and improve their data management strategies.
### Research questions

We aim to answer the following research questions:

1. How can SMEs check their GDPR compliance when using cloud services?
2. How can SMEs improve their GDPR compliance when using cloud services?
3. How can SMEs track their customers' data[1] in the cloud?
4. How can SMEs ensure privacy in the cloud by monitoring and auditing data access?

### Contributions

The main contribution of this article is the presentation of a tool to help SMEs understand and validate their compliance with GDPR. Specifically, this contribution encompasses the following notable sub-contributions. (1) We have developed the `GDPRValidator` tool to assist SMEs in achieving "the principle of accountability", under article 5[2], which requires that companies take responsibility for how personal data is processed. In addition, this principle requires companies to demonstrate GDPR compliance by documenting their procedures and routines. (2) This tool can make companies aware of the importance of "privacy-by-design", according to article 25. This GDPR article indicates that the controller has to implement technical and organizational measures both at the moment of determining the means of processing and during the actual processing. If companies do not comply with this principle, the work they have to do later to comply with GDPR can become overwhelming. (3) `GDPRValidator` enables economic savings for SMEs by helping them to audit and monitor their data activities with low cost. Furthermore, companies that process personal data on a large scale as part of their core business must engage a data protection officer *(DPO)*. The *DPO* is responsible for monitoring GDPR compliance as part of the organizational measures referred to in article 25, as well as training staff in compliance and conducting audits. *DPOs* can also benefit from it. (4) The `GDPRValidator` tool provides recommendations about how SMEs can track their customers' data to control the third parties that have accessed the data at a given moment.

The rest of the article is structured as follows (see Fig. 1 for the paper road map). The background is presented in "The general data protection regulation (GDPR)", and "Related work" reviews related works. The methodology that the tool supports is described in "Methodology", while "GDPR compliance validation" is devoted to the GDPR compliance process. Details of the implementation of our tool are described in "`GDPRValidator`", and its validation is presented in "Validation". Finally, "Conclusions" contains our conclusions and some lines for future research.

## THE GENERAL DATA PROTECTION REGULATION (GDPR)

The General Data Protection Regulation (GDPR) (*GDPR, 2016*) came into effect on May 25, 2018, to harmonize data protection laws across all EU member states. GDPR was adopted in 2016 to replace the Data Protection Directive, which was introduced in 1995 to align data protection standards in the EU, making internal and cross-border data transfers easier. GDPR not only harmonizes data protection rules but also creates legal certainty and

---

[1] This research is concerned with both SME customers' and employees' data. However, for simplicity, throughout this paper, we will refer to the personal data of customers and employees of a company only indicating customers' data.

[2] For short, we refer to the articles of GDPR only by their number.

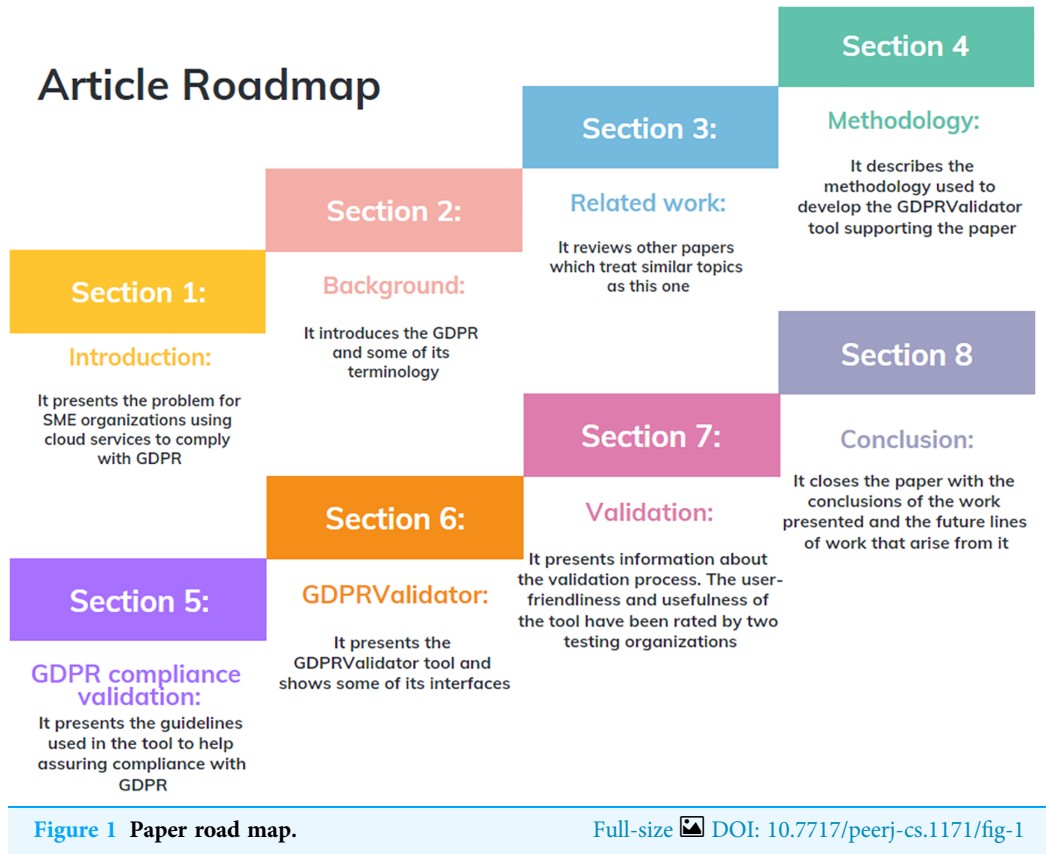

**Figure 1** Paper road map.                              

removes possible barriers to the free flow of personal data, raising the level of privacy. This law governs the automatic processing of personal data (or sets of personal data) by anyone who processes or controls them.

GDPR divides the actors into three groups: data subjects, the people whose data is handled (for example, customers or site visitors); controllers, who determine the purposes and means of data processing; and processors, who process personal data on behalf of the controller. Everyone, including minors, is protected by GDPR. Furthermore, it applies to the processing of personal data of EU residents both inside and outside of the EU.

A company that handles the data of EU citizens must demonstrate compliance with GDPR and take the necessary technical and organizational measures to protect the data. GDPR establishes a purpose limitation, which ensures that personal data is collected for specific, explicit, and legitimate purposes.

Additionally, it includes a system engineering approach, called privacy-by-design. This approach is based on six fundamental principles that aim to integrate data protection into the design of new products and systems. Specifically, these principles are (*Langheinrich, 2001*; *Cavoukian, 2009*): (i) proactive and preventive, not reactive; (ii) privacy as default settings; (iii) privacy built into the design; (iv) full functionality—positive-sum, not zero-sum; (v) end-to-end security—full lifecycle protection; (vi) visibility, transparency, and respect for the privacy of users—make it user-centric.

**Table 1 Comparison of GDPRValidator with other privacy compliance tools.**

|  | Regulations | Purpose | Cloud | Data breaches | DPIA | Documents generation | Free |
|---|---|---|---|---|---|---|---|
| *Vanta* | GDPR<br>ISO 27001<br>HIPAA | General | N/A | Y | N | Y | N |
| *privIQ* | GDPR<br>CCPA<br>LGPD<br>others | General | N/A | Y | Y | Y | N |
| *LogicGate* | GDPR<br>CCPA | General | Y | Y | Y | Y | N |
| *Cookiebot* | GDPR<br>ePR<br>CCPA | Websites | N/A | N | N | Y | N |
| *Ketch* | GDPR<br>CPRA/CPPA<br>LGPD | Websites | Y | N/A | N | Y | N |
| *FACILITA 2.0* | GDPR | General | N/A | Y | Y | Y | Y |
| *Facilita—Emprende* | GDPR | General | N/A | Y | Y | Y | Y |
| *GDPRValidator* | GDPR | General | Y | Y | Y | Y | Y |

Another relevant aspect of GDPR refers to the data subject's rights designed to guarantee their privacy and data protection. These rights are the right to information, the right to access, the right to rectification, the right to suppression, the right to limitation of treatment, the right to data portability, the right to oppose and the right to avoid automated decision-making.

Any breach in personal data must be reported to the data protection authorities and the data subject within 72 h of the company becoming aware of it, according to GDPR. There are different types of fines (less severe or more severe) depending on whether the notification is made on time or whether the data breach is caused by the negligence of the data controller or processor.

# RELATED WORK

In the literature we can find some tools to assist companies in becoming GDPR compliant. Table 1 shows a comparison of some of these tools. The column indicate the regulations considered in the tools, the general or specific purpose of the tools, whether they support compliance with cloud solutions, analysis of data breaches, Data Protection Impact Assessment (DPIA), whether the tools generate some documents to help in the compliance process, and whether the tools are free. Software compliance has been a relevant research area for decades (*Nair et al., 2014*). Many software systems have to comply with specific standards and regulations. Standards and regulations can cover different aspects (*de la Vara, Ruiz & Blondelle, 2021*), such as process or product characteristics or specific quality

concerns, such as safety, security, or privacy. Multiple stakeholders, from regulatory authorities to customers, can request and monitor the compliance process. Compliance management can be a complex process in practice because of various challenges (*Nair et al., 2015*), *e.g.*, having to comply with a large number of criteria, managing and showing compliance cost-effectively, or suitably understanding and following the standards and regulations. Compliance needs depend on the domain and features of the software system under consideration.

According to *Ryan, Crane & Brennan (2021)*, organizations face challenges in meeting GDPR compliance obligations despite having access to a large number of tools intended to assist them. They conclude that the ability of GDPR tools to demonstrate compliance has significant gaps: lack of interoperability, published methodologies or insufficient evidence supporting their validity and utility are the primary problems identified. *Buckley, Arner & Barberis (2016)* discuss *RegTech*, a framework for automating, monitoring, and reporting regulatory compliance in financial markets more efficiently and at low costs. The authors also analyze how *RegTech* systems can be applied to improve GDPR compliance by determining the key success factors.

As examples of general tools, *Vanta* (*Vanta, 2021*), *PrivIQ* (*PrivIQ, 2022*) and *LogicGate* (*LogicGate, 2022*) support general assessments of compliance with GDPR, as well as with other regulations, such as California Consumer Privacy Act (CCPA), California Privacy Rights Act (CPRA), ISO 27001, Health Insurance Portability and Accountability Act (HIPAA), Brazil's Lei Geral de Proteção de Dados (LGPD) and others. However, these tools typically do not address the specific compliance needs of cloud settings in detail. In addition, as they are not free, their use does not contribute to economic saving in SMEs.

There are other tools focused on analyzing GDPR compliance of websites. For instance, Cookiebot (*Cookiebot by Usercentrics, 2022*) and Ketch (*Ketch, 2022*). Cookiebot checks whether a website's use of cookies and online tracking is compliant with GDPR and the ePrivacy Directive (*ePR*) (*European Commission & Directorate-General for the Information Society & Media, 2015*). Ketch is focused on cookie analysis, consent, and DSR (Data Subject Request) requests for website and mobile devices. It generates reports containing data collection, exchange, access, and use in all its internal systems.

The Spanish Data Protection Agency (AEPD) provides several tools to help companies in the treatment of personal data, such as *FACILITA 2.0* (*Spanish Data Protection Agency, 2022a*) and Facilita Emprende (*Spanish Data Protection Agency, 2022b*). Nonetheless, these tools are generic and do not provide specific support for companies offering their services in the cloud. *FACILITA 2.0* helps companies to process personal data with little risk of non-compliance. This tool also provides some initial support documents aimed at simplifying the understanding of the obligations of the regulation. However, this tool cannot be used for processing involving high risks to the rights and freedoms of individuals, such as health data or massive data processing, among others. Facilita Emprende is designed for entrepreneurs and new companies (less than 10 years old) that have high growth expectations and conduct treatments that are based on emerging technologies. In this case, data processing is sometimes carried out in the context of the

management of a company's business activities or as a result of its services to third parties, which is not considered low risk.

It is also easy to find publications that propose other general solutions for GDPR compliance in the form of, for example, checklists (*Information Commisioner's Office, 2022*), frameworks (*Brodin, 2019*), processes (*Lioudakis et al., 2019*), or models (*Torre et al., 2021*). However, the specific needs of SMEs and cloud services are not adequately addressed, or if they are, these needs are not sufficiently addressed. For example, the suggested checklists can be too short for a thorough GDPR compliance assessment.

Regarding other approaches for compliance management, generic support that applies to multiple standards and systems has been proposed. This support includes metamodels (*de la Vara et al., 2022*), processes to take advantage of existing languages, such as UML (*Panesar-Walawege, Sabetzadeh & Briand, 2013*), UML profiles (*Giachetti, Marín & De La Vara, 2020*) and SPEM (*Ardila, Gallina & Muram, 2018*), and concrete tool support through the development of novel tools or the integration of various tools, *e.g.*, OpenCert (*de la Vara, Ruiz & Blondelle, 2021*). Most of this prior work was not developed within the scope of privacy compliance or GDPR compliance, but for different quality properties and specific system types, *e.g.*, safety-critical systems. Regardless of the system type or the standards involved, compliance management in different situations shares characteristics and core principles, such as the need to show that certain requirements are fulfilled. However, for application in GDPR compliance by SMEs that offer cloud services, the way to adapt these generic approaches would need to be established. Furthermore, such adaptations typically result in support that it is not the most suitable because the approaches initially targeted different contexts and purposes. Taking OpenCert as an example, its scope is much wider than what we aim for in this article, and its use for the case that we are dealing with would be more complex and inefficient.

Finally, notice that most companies have migrated their services to the cloud to be more competitive. The cloud allows them to reach a wider market and take advantage of cloud services. When these companies offer services in the EU or outside the EU, they must comply with GDPR when handling the personal data of their clients or employees. Thus, in the scope of the compliance of cloud solutions with GDPR, prior work has analysed its needs (*e.g.*, *Russo et al. (2018)*) and addressed specific technical aspects. For example, *Barati et al. (2022)* present a container-based monitoring and auditing architecture for improving data privacy in cloud ecosystems. Blockchain technology is used to implement the architecture, which is compliant with GDPR obligations in tracking cloud provider activities. The system can be used to verify compliance with four GDPR obligations by cloud providers, namely, data protection, minimization, transfer, and storage. *Corrales, Jurčys & Kousiouris (2019)* propose the use of smart contracts to fine-tune and embed GDPR requirements in the early architectural design. The authors develop a pseudo-code language for these contracts by taking into account legal and technical needs, which could be applied in blockchain contexts. Blockchain has also been considered as a technology to facilitate GDPR compliance by *Aujla et al. (2020)*, who propose an architecture for compliance provisioning, monitoring, verification, and enforcement. For GDPR compliance, as well as security assurance, *Rios et al. (2019)* present the DevOps framework

to support the design, deployment and operation of cloud systems. The framework considers the privacy and security controls necessary to ensure transparency to end-users, third parties involved in service provision, and authorities. The LogicGate (*LogicGate, 2022*) platform also allows monitoring cloud systems for operational performance, availability, and security. LogicGate complies with applicable data privacy laws, including GDPR. Nonetheless, it does not focus on SMEs' and DPOs' needs.

## METHODOLOGY

In this section, we describe the methodology used in our tool, namely `GDPRValidator`, to help SMEs in the process of being GDPR compliant. A running example is used to illustrate the whole methodology, which is presented in "Running example". Afterwards, "Methodology" explains the methodology.

### Running example

GestF is a small consultancy that offers its services in the cloud and operates in Spain. As part of this process, it processes the personal data of its customers and employees, such as salary information, electronic certificates, or other economic data. One of the services offered by GestF is the preparation of tax returns. Clients' and employees' data is stored both locally and in the cloud by this consultancy.

GestF has signed a services level agreement (SLA) with the cloud provider to store its customers' data. This contract must explicitly indicate the authorized third parties (recipients) to access the data. In this case, the SLA specifies Santander and ING as authorized third parties. GestF must comply with GDPR as it operates within the EU and processes EU citizens' data. In turn, it must identify where the information is stored, who accesses it, inform customers of their rights, request consent when a third party wants to use the data, and perform a Data Protection Impact Assessment (DPIA) when its data management involves "a high risk" to other people's personal information. However, GestF is not familiar with GDPR and does not understand the steps and documentation required to comply with it. As a consequence, it seems it needs to hire a legal expert to help it in this process, but it does not have a sufficiently large budget. According to GDPR, when a third party wants to access the data but has not been authorized by the SLA (condition 1), they must request the "consent" of the interested party (GestF customer) to access it. Then, this third party can access the data when the client gives the "consent" (condition 2).

Lacking proper knowledge of GDPR, GestF does not have control over whether these conditions are met. Any third party could access the data it handles, which represents a risk to privacy. For example, a consulting firm called Advisory could access the data of one of the company's clients, whose name is José Castro, without being authorized in the SLA or asking him for consent. José Castro's data include bank records, which is why the breach occurred.

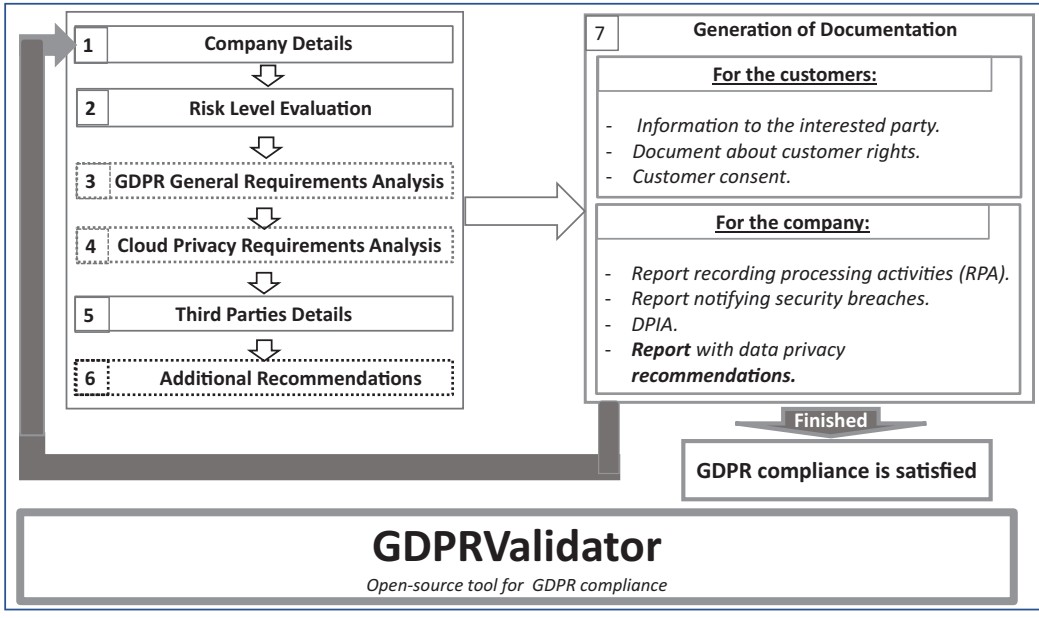

**Figure 2** Methodology for GDPR compliance validator.

## Methodology

Figure 2 illustrates the methodology, which consists of seven phases. Note that the phases framed by dotted lines are optional for SMEs. In the initial methodology phases (left-hand side of figure), the SMEs identify the risk they face when handling personal data. Then, they assess GDPR compliance by answering several questions. SMEs obtain various documents (on the right) that allow them to understand the GDPR requirements and how to become GDPR compliant.

The different phases of this methodology are as follows:

1. Company Details. In this phase, the SMEs fill out their details, such as business name and address, which will be used to provide the generated documentation with this information.

2. Risk Level Evaluation. During this initial phase, the level of data management risk is analyzed. When this risk assessment concludes that there is a high risk for the data handled by the company, the guidelines to carry out a Data Protection Impact Assessment (DPIA) are given. Under GDPR, you must conduct a DPIA whenever you begin a new project that involves a high risk to someone else's personal information.

3. GDPR General Requirements Analysis. GDPR.EU proposes a checklist (*GDPR.EU, 2022d*) to help controllers become more secure, protect customer data, and avoid costly fines for noncompliance. During this phase, we use this general GDPR checklist to evaluate SMEs' compliance with GDPR requirements by using a set of questions based on it.

4. Cloud Privacy Requirements Analysis. This phase focuses on evaluating the privacy requirements needed by SMEs to comply with GDPR. The SME has to answer a cloud-

related set of questions. The two previous phases allow the SME to analyse its compliance with the general and cloud-related GDPR requirements. These phases are independent, so the companies can answer the questions of each step independently or answer both parts. This feature makes it easier for the company to use the tool since maybe it had already obtained an initial assessment of its GDPR compliance and has been working, for instance, on the general recommendations. Then, it may only want to assess the cloud-related requirements to obtain the recommendations related to that part. This feature makes `GDPRValidator` much more efficient to use and makes it easier for SMEs to use.

5. Third Parties Details. In this phase, the SMEs can fill out the details of all those companies (third parties) that wish to access the customers' data before they give the "consent". For each of these companies, the SMEs must enter the following information: company category, the purpose for access, action, and duration of storage (in the case of data copy).

6. Additional Recommendations. Before generating the documentation, the company can also indicate whether or not it wants to obtain the additional recommendations offered by the tool to improve its GDPR compliance.

7. Generation of Documentation. In this phase, the `GDPRValidator` tool generates the corresponding SME documentation according to the result of the risk evaluation and the answers given by the company to the questions defined in the previous phases. The company receives several documents from this phase. These documents allow the company to adjust its security policies to comply with GDPR. These include those documents that a company should provide to its customers and those that it should use for internal control in GDPR compliance. The documents are available in .doc format. For its customers, the company obtains three documents:

- A document containing the information specified in article 13. It includes information that the controller must provide to the customer. For instance, some of this information is the controller identity and contact details or the data protection officer, if any.

- A report explaining the rights of the customers (*Altorbaq, Blix & Sörman, 2017*) and how they may exercise those rights, chapter 3, articles 12–23.

- A document that enables obtaining of the consent of the customer allowing access to its data.

 For the internal control of the company, the tool generates the following documents:

- A report of processing activities (RPA) according to article 30. In GDPR, this document is mandatory for large companies, but it is also recommended for SMEs.

- A document to inform the supervisory authorities of security breaches, as outlined in article 33.5. In the case of a personal data breach, the controller must inform the supervisory authority competent without undue delay and, where feasible, not later than 72 h after becoming aware of it, as article 55 stipulates. If there is a personal data breach, the controller must document the facts relating to the breach, its effects, and the

remedial actions taken. This report must include an explanation if it is not sent within 72 h.

- A document containing the data protection impact assessment (DPIA), in accordance with article 35. In cases of automated data evaluation of personal aspects, such as profiling, processing special categories of data on a large scale (article 9), criminal convictions and offenses (article 10), or systematic monitoring in a publicly accessible area on a large scale, the document is required.
- A report with data security recommendations. In this document, the company will find some helpful recommendations for GDPR compliance.

After completing all these phases, and taking into account the actions performed to solve the problems encountered, the SME can repeat the whole process to determine the GDPR compliance. Finally, it is important to remark that once the tool execution finishes, the data provided by the company during the process is destroyed to guarantee the company's privacy.

## GDPR COMPLIANCE VALIDATION

This section presents the guidelines used in our tool. An SME can use `GDPRValidator` to assess its GDPR compliance by checking whether these guidelines are followed. As a result, it provides suggestions to help the company comply with them.

We divide the following description into five sections. The first section describes the risk level evaluation, corresponding to methodology phase 2 (Fig. 2), in which a categorization of the risks and the importance of performing the DPIA are explained. The second section lists the general guidelines for the data controller, based on the GDPR checklist (*GDPR.EU, 2022d*), methodology phase 3. The third section focuses on cloud security requirements, methodology phase 4. In the fourth section, we present several additional suggestions to help companies comply with GDPR, methodology phase 6. Finally, the last section summarizes the results.

### Risk levels evaluation

Companies must conduct a DPIA when there is a high risk to people's rights and freedoms, according to article 35. GDPR takes into account the principle of accountability. The importance of this principle lies in the fact that it forces data controllers to identify, assess, and manage risks to these rights and freedoms. During this process, it is essential to be able to identify and evaluate risks in the field of data protection based on a common view of what constitutes a risk (*Demetzou, 2019*). The management of risk follows several steps. Firstly, the categorization of the data handled and the processing activities performed on them. Then, these risks are evaluated, prioritized, and countermeasures and mitigation measures are taken. The final step includes documenting and monitoring (*Martin & Kung, 2018*).

In this article, we focus on categorizing data that the companies handle and on evaluating the risk level that this management involves. When the risk is "high", the `GDPRValidator` tool recommends performing a DPIA. However, we do not generate a

risk analysis since this task is not the focus of the current work, but we will consider it in future work. "Categorization of risks" defines the categories of data that imply a "high" risk, while "Data protection impact assessment" explains the DPIA concept.

### Categorization of risks

According to recital 77 of GDPR, codes of conduct, certifications, guidelines, or information issued by the European Data Protection Board (article 68) or a data protection officer's guidance are acceptable methods for implementing measures and demonstrating compliance by the controller or processor. Risk assessment is based on the risk's origin, nature, likelihood, and severity, and where appropriate, it is also vital to define sufficient measures to address a given risk. In this case, GDPRValidator recommends performing a DPIA for a deeper analysis.

A categorization of the risks considered as having a high or very high impact is as follows:

- Data processing risks in general.

1. Special categories of personal data according to article 9.
2. Data relating to criminal convictions and offenses according to article 10.

- Data processing risks in cloud environments.

1. When multiple infrastructure and software providers are involved in a cloud-hosted application, the risk of data breaches can be high, since several third parties can try to access the data for different purposes.
2. Cloud environments lack the necessary transparency in the access to the data of the data subjects. The company's customers (data subjects) should give their "informed consent" to demonstrate that they have consented to their data processing, article 7. And they shall have the right to withdraw their "consent" at any time.
3. One of the most distinguishing features of cloud computing is data mobility, which refers to the ability to share data among cloud services and recover this data if a user stops using cloud computing services. It will be necessary to track the data to remove all copies.

### Data protection impact assessment

This section discusses the importance of performing a Data Protection Impact Assessment (DPIA) for companies to show compliance with GDPR (article 35). GDPR requires this process before starting a new project that involves a high risk to others' personal information or before beginning any high-risk data processing activity. According to the "protection by design" principle, this is one of the most important ways to demonstrate to authorities that an organization complies with GDPR. GDPR does not exempt businesses with fewer than 250 employees, as commonly believed. In (*GDPR.EU, 2022a*) the reader can find some examples of the types of conditions that would require a DPIA.

In this article, we consider companies that meet some of these criteria: they employ new technologies, they can track people's location or behavior, or they may use data processing to make automated decisions about people that may have legal consequences. According to GDPR, companies with these characteristics should develop a DPIA. To give SMEs a guide on how to perform a DPIA, we use a template provided by the Information Commissioner's Office of the United Kingdom (*Information Commissioner's Office, 2019*), which is responsible for enforcing GDPR in that country.

According to GDPR, a DPIA should include the following elements:

1. A description of the intended processing operations and their purposes, including, where applicable, the legitimate interest of the controller.
2. Concerning the purposes, an assessment of the necessity and proportionality of the processing operations.
3. The measures taken to minimize the risks to the rights and freedoms of data subjects.

Following article 36, in those cases in which a DPIA indicates that the processing will result in high risk and the controller does not take measures to mitigate the risk, we recommend that the controller consult the supervisory authority before processing.

## Validation of general GDPR compliance guidelines

GDPR provides a checklist (*GDPR.EU, 2022d*) for data controllers to help maintain a secure organization, protect customers' data, and avoid costly fines for non-compliance with GDPR. This checklist consists of questions about how these companies apply GDPR. If the company's response to some of the questions suggests that the company is not compliant with GDPR, then the necessary recommendations are added by GDPRValidator to the report generated for the company.

We include only a few of these questions here, and we use the running example to illustrate the suggestions that will be added to the report generated for the GestF consultancy, including data privacy recommendations. The complete list is available in Appendix A. The questions are divided into five categories based on the data security issue addressed.

**1. Lawful basis and transparency.**

(a) *Does the company conduct an information audit to determine what information is processed and who has access to it?*

If GestF answers no to this question, then the following suggestion is added: The most effective way to demonstrate GDPR compliance is to conduct a data protection impact assessment (DPIA), article 35. Among other things, the DPIA must include the purpose of the processing, the types of data you process, who has access to it in the organization, any third parties (and their location) who have access, how you plan to protect the data (such as encryption), and when you plan to erase it (if possible). The CSP (Cloud Service

Provider) should also provide your company with all this data access information. Therefore, the CSP should prepare its DPIA to keep track of all cloud accesses. This issue can be agreed upon when signing the SLA with the CSP. Your company will need to ask the CSP for their DPIA to complete the company's DPIA.

The `GDPRValidator` tool allows companies to assess whether they need a Data Protection Impact Assessment (*GDPR.EU, 2022b*) (DPIA) (see "Data protection impact assessment") and generates a draft of it.

**2. Data security.**

**(a) *When a data breach occurs, does your company have a procedure for contacting the authorities and your data subjects?***

If GestF answers no to this question, then the following suggestion is added: Here, we provide a list of the EU member states' supervisory authorities. And we include in the recommendation report the following: According to article 34, you must also notify the supervisory authority and your customers immediately, within 72 h, if there is a data breach; unless the breach is unlikely to put them at risk, for instance, if the data stolen is encrypted.

**3. Accountability and governance.**

**(a) *Has your company signed a data processing agreement with any third parties who process personal data on your behalf?***

If GestF answers no to this question, then the following suggestion is added: For GDPR compliance, almost all services require a standard data processing agreement, which includes any third party services that handle your data subjects' personal information.

A template for this agreement, specifying the rights and obligations of each party, is available at the following link: https://gdpr.eu/data-processing-agreement/, which must be visible on your website. It is imperative to only work with third parties who can provide adequate data protection guarantees. Thus, the SLA with the CSP must specify the list of permitted third parties.

**4. Privacy rights.**

**(a) *Can your customers easily access all the information you have about them?***

If GestF answers no to this question, then the following suggestion is added: First, your organization should ensure that the customer requesting the data is verified. Your customers have the right to know what personal information your company has about them, how it is used, how long it is stored, and why it is stored (article 15). You must respond to these requests within one month (article 16). This information is sent for free the first time it is requested, but subsequent copies may have a cost.

**5. Third Parties.** `GDPRValidator` also requests the details of all those companies (third parties) that wish to access your customer's data before the "consent".

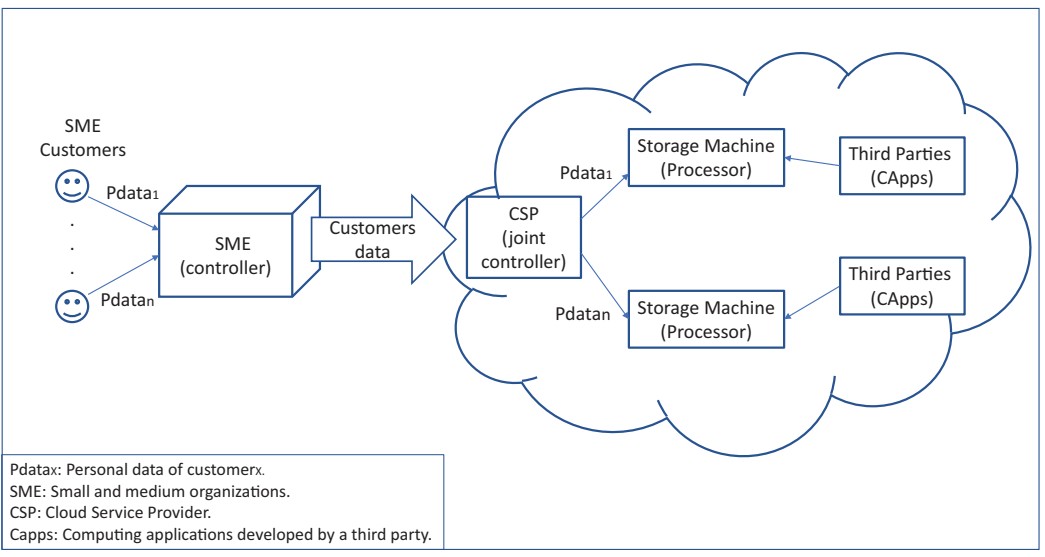

**Figure 3 GPDR roles interacting in a cloud system.**

**(a)** *Do you have third parties requesting "consent" to access your customers' data?* Note that for each of these companies `GDPRValidator` allows you to enter the following information: company category, the purpose for access, action, duration of storage, in the case of data being copied.

If GestF answers yes to this question, then the following suggestion is added: Your company must have the identification of those companies that wish to access your customer's data before the "consent" is given, as well as the purpose of the access, the action to be carried out on the data, and if this data is to be copied, the duration of storage of that copy.

## Validation of cloud-related GDPR compliance guidelines

There are many benefits of storing data in the cloud, such as lower data storage and management costs. There is also often less chance of data loss. However, most companies that contract cloud storage services have raised concerns about potentially violating GDPR compliance requirements by storing data in these systems. This concern could stem from a perceived loss of control once the data is in the cloud. A set of questions is provided in this section to help SMEs determine whether their cloud provider complies with GDPR.

Figure 3 shows how the GDPR roles interact in a cloud system. We can see that the SME is the controller of the customers' data, being responsible for the treatment of this data. Note that the company's customers must sign an SLA with the company regarding their data treatment. Then, the SME can hire cloud services to store its customers' data on some storage machine in the cloud system. The SME also signs an SLA with the CSP, which allows it to be joint controller (article 26). Thus, each cloud storage machine, on which the data is stored becomes a data processor and is responsible for processing the data

(article 28). Then, any third party that obtains the corresponding "consent" should be able to access the stored data. The controller and joint controller shall implement the appropriate technical and organizational measures to ensure, and be able to demonstrate, that processing is performed in accordance with this regulation (articles 24 and 26).

The main questions that allow `GDPRValidator` to validate GDPR compliance are inspired by *Deloitte (2022)*, *Combell (2020)*. Below we list only a few of these questions and the suggestions that will be added to the report generated for GestF, including data privacy recommendations, using the running example as an illustration. The complete list is available in Appendix A.

**1. Contracts.**

**(a) *Has the CSP signed an SLA for data processing with any third parties who access/process personal data?***

If GestF answers no to this question, then the following suggestion is added: For GDPR compliance, almost all services require a standard data processing agreement, which includes any third party services that handle your data subjects' personal information. It is imperative to only work with third parties who can provide adequate data protection guarantees. This suggestion complements the related one in point 3 under *Accountability and governance* in the general GDPR compliance guidelines.

**2. Duration.**

**(a) *Does your company establish a date for data retention?***

If GestF answers no to this question, then the following suggestion is added: The storage of data should not exceed the necessary storage time. The SLA signed with the CSP must specify the processing duration. After the retention period has expired, all data, including copies, must be deleted.

**3. Information.**

**(a) *Does your company specify a procedure to notify when a breach is detected?***

If GestF answers no to this question, then the following suggestion is added: GDPR in the article 4 determines what is considered a breach in the collection, retention, and processing of data: "the accidental or unlawful destruction, loss, alteration, unauthorized disclosure of, or access to, personal data transmitted, stored or otherwise processed". Then, the company must specify the procedure that it follows to notify when a breach is detected, for instance, in the event of theft or a leak. Before a breach is reported in the media, the CSP should inform your company, and your company should notify its customers and supervisory authorities. According to article 36, the controller must report any breach of personal data to the supervisory authority competent under article 55 within 72 h of becoming aware of it, unless the breach is unlikely to result in a risk to the rights and freedoms of natural persons. This suggestion complements the related one in point 2a under *Data security* in the general GDPR compliance guidelines.

**4. Privacy rights.**

**(a) *Can your company access and delete the customers' data from the cloud?***

If GestF answers no to this question, then the following suggestion is added: Your company has the right to access its customers' data and have them deleted, the right to access (article 15), and the right to erasure ("to be forgotten", article 17). The CSP must facilitate this process by making the data available to your company and your customers in a structured format. It is also important to consider data backups when deleting customer data.

**5. Data Protection Impact Assessment (DPIA).**

**(a) *Does the CSP have a Protection Impact Assessment (DPIA) that determines the risks associated with cloud hosting or services?***

If GestF answers no to this question, then the following suggestion is added: The CSP, as a joint data controller, should also have a DPIA (see "Data protection impact assessment") that determines the risks associated with cloud hosting or services. It is recommendable, but not mandatory. The CSP must send this DPIA to your company every time that it is updated. This suggestion complements the related one in point 1a under *Lawful basis and transparency* in the general GDPR compliance guidelines.

**6. Third Parties.**

**(a) *Do you have third parties in the cloud requesting "consent" to access your customers' data that are not included as recipients in your contracts?***

In this case, only when GestF answers yes to this question is the following suggestion added: Your company must have the identification of those companies that wish to access your customer's data before the "consent" is given, as well as the purpose of the access, the action to be carried out on the data and, in the case of copying this data, the duration of storage of that copy.

## Additional Sugestions to GDPR compliance

In this section, we offer some additional recommendations based on our research to assist companies in achieving greater GDPR compliance. It is optional for companies to obtain this information. This list aims to offer additional assistance to SMEs in order to make compliance with EU GDPR as easy as possible. It is added at the end of the recommendations report.

1. Sticky policies. Your organization should create security policies associated with data in the form of sticky policies (*Pearson & Casassa-Mont, 2011*). A sticky policy is a machine-readable policy attached to data in order to define the permitted use and obligations for the data. These policies travel with the data, allowing users to enhance control over their personal information. A portion of the sticky policy information comes from the contract

with the CSP. This suggestion complements the one offered to the data security question 2d in the checklist in the general GDPR compliance guidelines.

2. Control accesses. The CSP should limit access to data. In the SLA between the company and the CSP, the company should establish the recipients[3], *i.e.*, the third parties authorized to access its customers' data and for what purpose. This suggestion is related to the question in the checklist in accountability and governance in the general GDPR compliance guidelines, and questions 1a and 1b in the checklist in the cloud-related GDPR compliance guidelines.

Therefore, we propose to include this information in the permission and purpose fields of the sticky policy associated with the data. For example, Google Cloud provides a list of the processors permitted to access its data, following this recommendation, as seen at https://cloud.google.com/privacy/gdpr.

3. Tracking data accesses. The presence of third parties can increase risk, as explained in "Categorization of risks". We recommend that the CSP, as joint controller, identify third parties before the company adopts a cloud service, ensuring that their responsibilities are addressed in the contract. In this way, only third parties who obtain the explicit "consent" can access the data. For this purpose, we propose to add a field, called *accessHistory*, to the proposed security sticky policy associated with the data that allows the recording of the data accesses of third parties, that is, allows the tracking of the data.

4. Encryption and edge computing. We recommend encrypting the data locally (*Duncan, 2018*). This recommendation is based on a report by the European Union Agency for Network and Information Security (ENISA) (*Izenpe, Primekey & Enisa, 2013*). Data encryption and decryption keys must be stored by the data controller (your company) since both keys and data must remain under your control, not in the cloud. For this purpose, one solution could be the use of edge computing technology. In this way, outsourced data storage on remote clouds is practical and relatively safe, as long as only the data owner, not the cloud service, holds the decryption keys.

Therefore, considering these additional suggestions, we propose that the sticky policies associated with the data should have the following fields:

$$DS\_SP = \{\{permission : list\ of\ (third\ party,\ action)\},$$
$$\{owner : data\ subject\},\ \{purpose : list\ of\ string\},$$
$$\{controller : [controller,\ joint\ controller]\},$$
$$\{accessHistory : list\ of\ (third\ party,\ purpose,\ action)\}\}$$

The sticky policy will be attached to the data and travel with it wherever it is stored. In the permission field, the list of third parties authorized to access this data is defined, together with for what action. This action can be a reading, writing, or copying of the data. The *owner* field establishes the data subject. The purpose field indicates the list of purposes permitted for the data access. It could be for statistical or other purposes. In addition, it includes the data controller. Furthermore, the *accessHistory* field allows tracking of the data, tracking who has accessed the data, what they did, and for what purpose.

[3] According to article 4, the term "recipient" means a natural or legal person, a public authority, an agency, or another entity to which the personal data are disclosed.

In the running example, the consultancy Advisory wanted to read "José Castro's" data, but it is not included in the initial SLAs, so it must request *"José Castro's informed consent"* to read the data. Initially, the permanent policy for this data (*JC_SP*) is:

$$JC\_SP = \{\{permission : [(GestF, *), (SB, read), (ING, read)\}], \{owner : JC\},$$
$$\{purpose : [taxes, statistical]\}, \{controller : [GestF, CSP]\},$$
$$\{accessHistory : [(SB, taxes, read)]\}\}$$

Therefore, `GestF, SB, ING` can read the data. The owner of this data is the `JC`. The list of purposes has two types: statistical and tax. The controller and joint controller are GestF and the CSP. Finally, the *accessHistory* field allows us to track this data. Based on this field, the entity (`SB`) has read the data for taxes purposes. Once *JC* consents to the access, Advisory can read the data.

The JC data sticky policy is modified as follows:

$$JC\_SP = \{\{permission : [(Advisory, read), (GestF, *), (SB, read), (ING, read)],$$
$$\{owner : JC\}, \{purpose : [taxes, statistical]\},$$
$$\{controller : [GestF, CSP]\},$$
$$\{accessHistory : [(Advisory, taxes, read), (SB, taxes, read)]\}\}$$

According to the permission field, the Advisory consultant has reading permission. The *accessHistory* field indicates that both the Advisory and the SB have already read the data for tax purposes. Therefore, this last field would allow the tracking of *JC's* data.

## Results

This section describes the content of all the documents generated by the `GDPRValidator` tool to help SMEs to be GDPR compliant. In "Documentation for the customers", we describe the documentation to send to the company's clients, focusing mainly on the exercise of their rights. A description of the documents generated for the internal control of SME GDPR compliance is given in "Documentation for the company".

### *Documentation for the customers*

The documentation generated by `GDPRValidator` for the SME's customers is the following:

1. Information to the interested party. In accordance with article 13, this document contains the information that must be provided to customers when their data is collected. These documents provide, among other things, the following information: the identity and contact information of the controller and the data protection officer, if applicable; the purposes of the processing; the recipients; and, the storage period.
2. Document about customer rights. It contains information on how customers can exercise their rights, that is, the right to access, rectification, erasure, restriction of processing, data portability, and to object (Chapter 3, articles 12–23 (rights of the data subject)).

3. Customer consent. This document allows the company to obtain the consent of the customer for allowing access to its data. The company will use this document to request the customer's express consent when a third party wants to access its data. This document identifies the third party, the purpose of access, and what action the third party wishes to be able to perform.

### Documentation for the company

The documentation generated by `GDPRValidator` for the SME, as data controller, is the following:

1. Monitoring. The CSP should keep a record of processing activities (RPA) of third parties who access the data and how they use it (article 30). Records of processing activities must include information about data processing, such as data categories, the group of data subjects, the purpose of the processing, and the data recipients. This documentation must be made available to authorities upon request. However, organizations with fewer than 250 employees are exempt from maintaining these records if the processing does not pose a risk to the data subject's rights, if no special categories of personal data are processed, or if it is occasional processing. In practice, this exemption is rarely applicable. The governments of the countries involved in data management impose fines on companies that fail to keep records of processing activities. The fines can be as high as 10 million euros or 2 percent of their annual revenue.

The monitoring process logs all data accesses. An SME should keep track of all accesses to its customers' data, both on-premises and in the cloud. SMEs should maintain a local record of all customer data accesses. At the cloud level, the CSP must also track data access in a similar document. These records must be kept current. CSPs must periodically send logs to SMEs or upon request. These logs provide the information used in RPA.

2. Security Violation Log Sheet. Under GDPR, all businesses must keep a record of all violations. This record should describe the event itself, the cause, the repercussions, the risk of future harm, the data affected, and the measures taken to mitigate the risk of further damage. `GDPRValidator` generates a template with the SME data to help it to recording data violations. We obtained this template from the following link: https://gdprwise.eu/data-breach/data-breach-incident-log-template.

3. DPIA. GDPR recommends that each company conduct a Data Protection Impact Assessment (DPIA) to prove compliance with GDPR (Article 35) (see "Data protection impact assessment"). `GDPRValidator` offers a template for this document including the SME's data details.

4. Report with recommendations. In this document we include a glossary of terms related to the main GDPR terms, such as data subject, controller, processor, consent or purpose. This glossary helps the company to better understand GDPR in its context. For example, if data subject is defined as the data owner, then the SME can identify its customers or employees as data subjects. Article 4 of the GDPR contains the definitions of each term, and this document also refers to it. This report includes recommendations for each of the

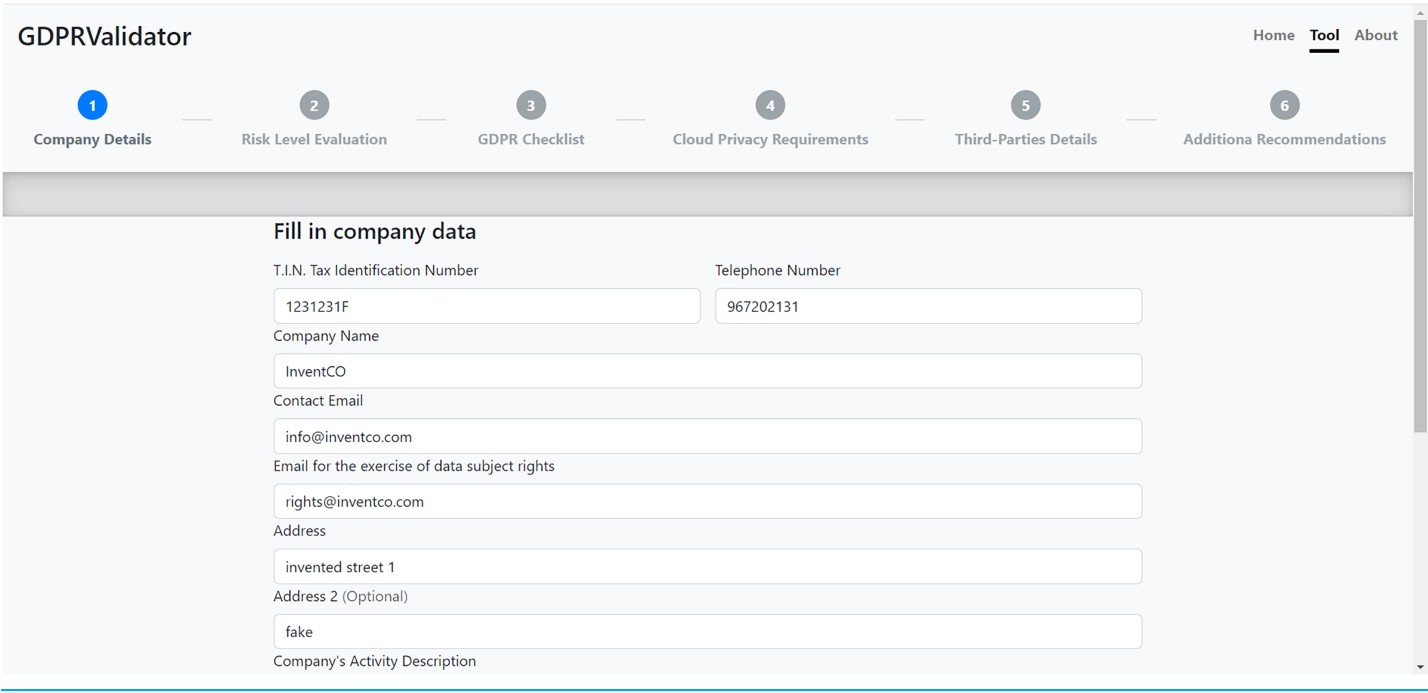

**Figure 4  Tool's company data form page.**               

weaknesses found in the SME's GDPR compliance, both general and related to the cloud. We also include additional recommendations at the end of this report if the company chooses to include them.

## GDPRVALIDATOR

In this section we present a description of the `GDPRValidator` tool. Web-based access is available at http://pluton.i3a.uclm.es:8080/, and it follows the methodology outlined in "Methodology". Professionals and businesses who handle personal data can benefit from this tool, as it allows them to evaluate the security of their processing of personal data. It also allows them to obtain a report with recommendations that help ensure data privacy in the context of GDPR. The company's data is destroyed after the tool's execution is complete and the set of documents is obtained. The resulting documents will only be valid if the answers to the questions are authentic and current.

### Design and interface

This web-app starts with a simple front page that has a brief description of the tool and a fragment of Fig. 2 including the sequence of steps in the main process. After these, a button takes the user to the beginning of the process, the SME's data form. `GDPRValidator` also includes an about page, reachable from any of the other ones, which expands on the information provided about the tool on the front page. Note that on each `GDPRValidator` page we have included a 6-step roadmap showing which stage the user (the SME or DPO) is at. Figure 4 shows the tool's page for registering the SME's details, *via* which the

company can fill in its information using the form fields needed for its identification. Note that this information is not stored, it is used only to adapt the resulting documents to the company in question. Nonetheless, they must review and complete them with more specific information.

Figure 5 shows both the interface screens of the `GDPRValidator`'s checklists pages, this is, the general GDPR guidelines, established by the official EU checklist (*GDPR.EU, 2022d*), and the cloud-related guidelines, introduced in "Validation of Cloud-related GDPR Compliance Guidelines". As can be seen, both consist of a series of yes/no questions, defined in "GDPR compliance validation". Each question also has a dropdown box that shows the resulting recommendation when selected. Before answering any questions, an option is provided to skip these pages, which are independent and optional steps in the tool's process.

Lastly, Fig. 6 shows the final page of the process. First, it states that the company using the tool must complete all the documents obtained to include any additional information regarding the data processing. The page provides a series of basic general guidelines for GDPR conformance. Then, it also contains a check box that SMEs can optionally mark if they want to receive the additional recommendations included in "Additional Sugestions to GDPR Compliance", in addition to those resulting from the previously explained checklists. As a final step, users will receive all the documentation generated by clicking the "FINISH" button.

## VALIDATION

The main goal of the validation is to study whether `GDPRValidator` is a feasible means of enabling SMEs that use cloud services to assess GDPR compliance. "Design" presents the design of the validation, and "Results" shows the results obtained. Finally, we discuss the validation and its results in "Discussion".

### Design

The validation was conducted with SMEs that use the `GDPRValidator` tool. First, we wanted to determine whether our tool is helpful for SMEs in understanding and complying with GDPR. Second, we wanted to evaluate its usability.

Two SMEs participated in the validation. The first one, Netberry Solutions S.L., (https://www.netberry.es/), is a company with 58 technology services employees and more than 20 years of experience in the ICT sector. The company has carried out over 1,500 projects for very different clients, including large companies, SMEs, public administrations, and startups. It handles both employee data and customer data, and for the storage of these data it uses cloud services. The other company is IeeAsesores S.L., (http://www.ieeasesores.com/), which provides integrated advice on tax, and labour issues accounting services. This company has nine employees and manages data from a considerable number of clients (about 2,000), and it stores part of the data, both about clients and employees, on-premise and the other part in the cloud.

The companies can access the tool at http://pluton.i3a.uclm.es:8080/. The evaluation was performed by the person responsible for data privacy in each company, and this

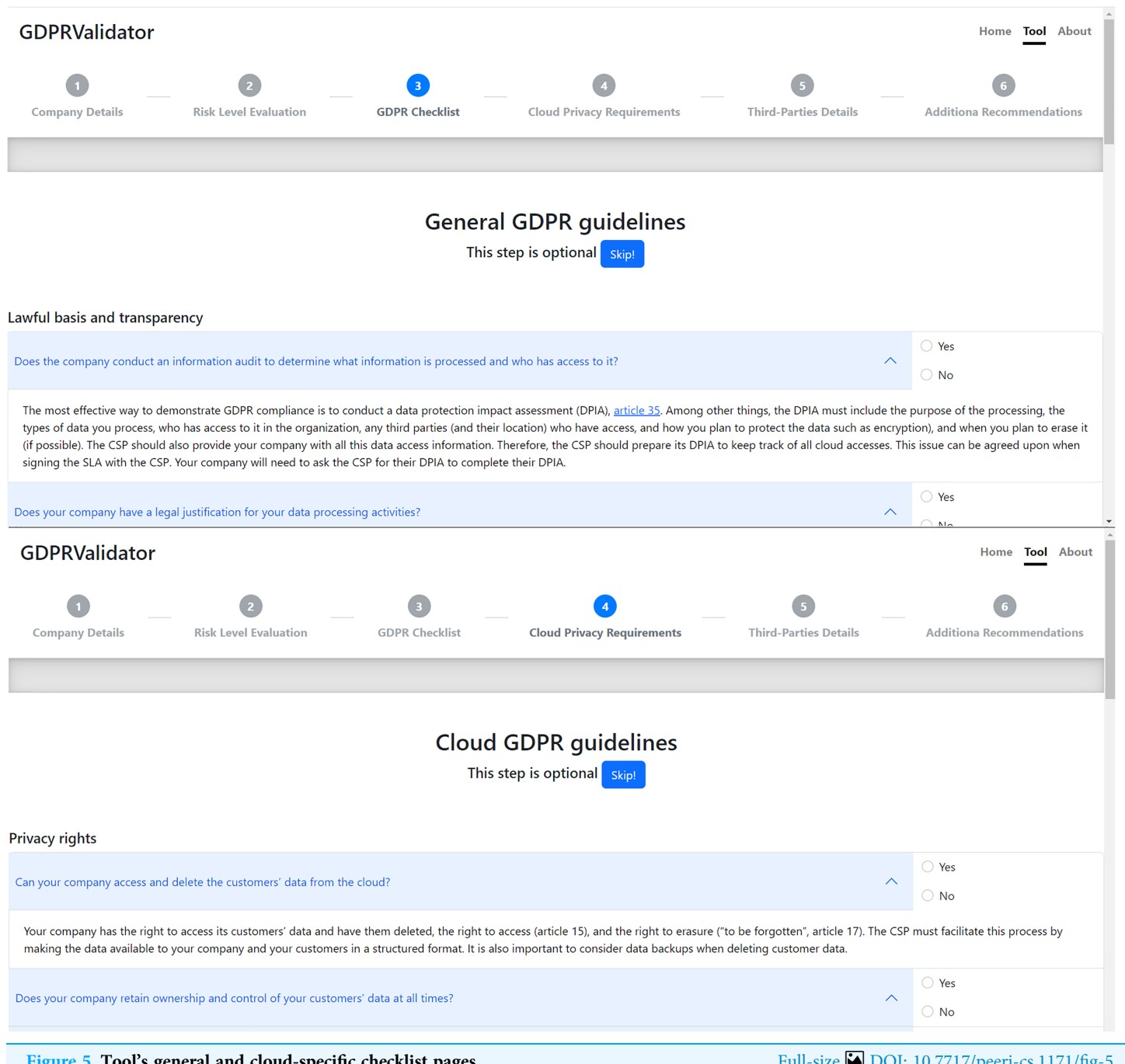

**Figure 5  Tool's general and cloud-specific checklist pages.**

person acted as the final user of the tool. In the case of *IeeAsesores* it was the person in charge of the company. At Netberry, there is a team that deals with data privacy. In both cases, the persons responsible for data privacy management answered the questions and the tool generated the documentation to aid them in complying with GDPR.

The `GDPRValidator` tool was evaluated using the well-known Technology Acceptance Model (TAM) (*Davis, Bagozzi & Warshaw, 1989*), which is based on the Theory of

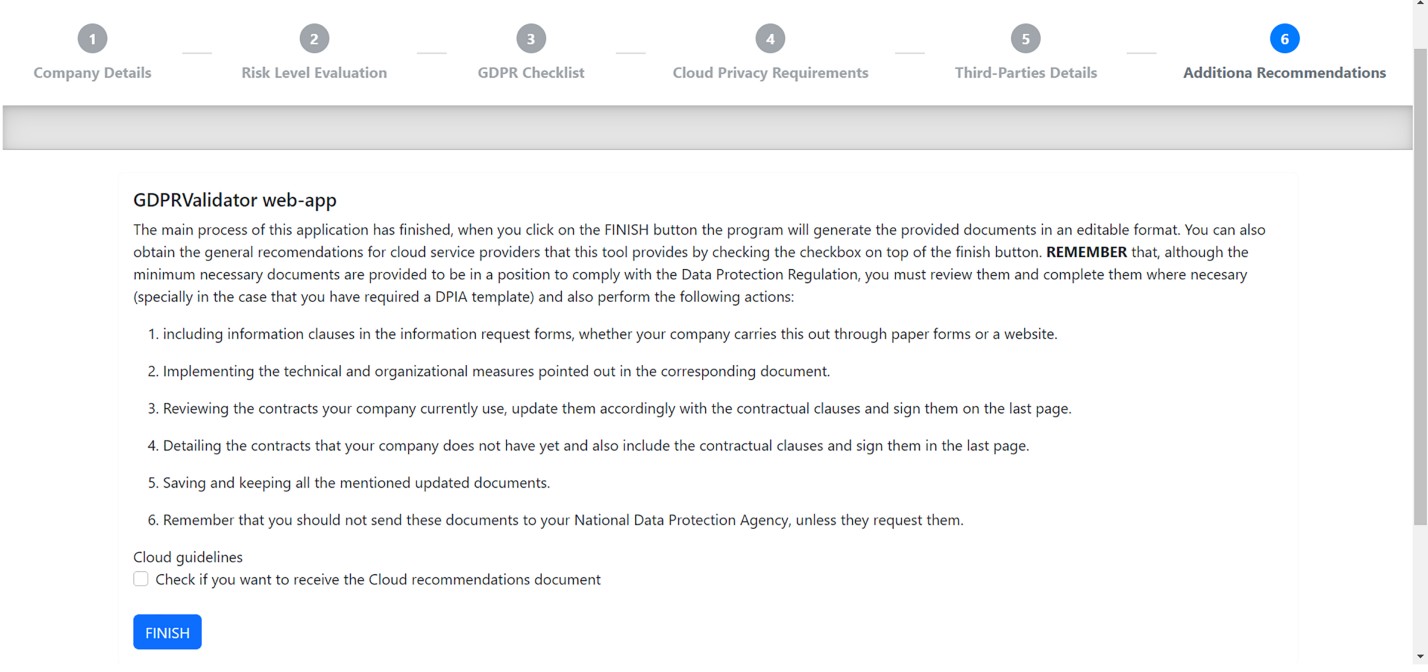

**Figure 6  Check for additional recommendations on the last `GDPRValidator` page.**

Reasoned Action (TRA) (*Fishbein & Ajzen, 1975*) and used to evaluate the acceptance of technology by users. The set of statements used (see Table 2), were adapted from (*Auer & Felderer, 2020*), which in turn adopts and adapts a set of reference TAM questions and statements, as other authors have also done. These statements measure the following variables: perceived usefulness, perceived ease of use, and future usage. We used a Likert scale to collect the participants' opinions, with values ranging from 1 (strongly disagree) to 7 (strongly agree)[4].

This study considered the following variables:

1. Perceived usefulness: the degree to which SMEs expect `GDPRValidator` to improve their GDPR compliance in handling customer personal data. The grade of this variable is determined by the six usefulness statements (*Ux*).

2. Perceived ease of use: the degree to which SMEs expect `GDPRValidator` to be easy to use. As with the previous variable, the grade of this variable depends on the grades from the six ease-of-use statements (*Ex*).

3. Future Use: the likelihood that SMEs will use `GDPRValidator` in the future. This variable's grade is based on two statements (*Sx*).

## Results

Table 3 presents the results for the variables under study. All the responses express agreement on the statements. The results can be summarized as follows:

1. Perceived usefulness. Both companies provided values above five (slightly agree) for all the statements, which means that the degree of usefulness perceived by SMEs of

[4] The values for the Likert scale are those following *Taherdoost (2019)*: (1) strongly disagree, (2) disagree, (3) slightly disagree, (4) neutral, neither disagree nor agree, (5) slightly agree, (6) agree, (7) strongly agree.

**Table 2 TAM scale items.**

| STATEMENTS | DESCRIPTION |
|---|---|
| **Perceived usefulness** | |
| U1 | Using GDPRValidator in my company would enable me to understand GDPR easily. |
| U2 | Using GDPRValidator improves the preparation of the documentation necessary to comply with GDPR. |
| U3 | Using GDPRValidator in the enterprise increases the privacy of the data managed. |
| U4 | Using GDPRValidator tool improves my effectiveness in complying with GDRP. |
| U5 | Using GDPRValidator makes it easier to ensure GDPR compliance. |
| U6 | I would find GDPRValidator useful for the company. |
| **Perceived ease of use** | |
| E1 | Learning to operate with GDPRValidator would be easy for me. |
| E2 | I would find it easy for my company to have GDPRValidator check for GDPR compliance. |
| E3 | My interaction with GDPRValidator would be clear and understandable. |
| E4 | It was easy to become skilful using GDPRValidator. |
| E5 | It is easy to remember how to perform tasks using GDPRValidator. |
| E6 | I would find GDPRValidator easy to use. |
| **Future usage** | |
| S1 | Assuming GDPRValidator would be available at my work, I predict I will use it regularly in the future. |
| S2 | I would prefer using GDPRValidator to other forms for understanding and dealing with the process of becoming GDPR compliant. |

**Table 3 Results in terms of the grades for GDPRValidator validation by the SMEs.**

| Company | Perceived usefulness | | | | | | Perceived ease of use | | | | | | Future usage | |
|---|---|---|---|---|---|---|---|---|---|---|---|---|---|---|
| | U1 | U2 | U3 | U4 | U5 | U6 | E1 | E2 | E3 | E4 | E5 | E6 | S1 | S2 |
| *Netberry* | 7 | 6 | 6 | 6 | 5 | 7 | 7 | 6 | 7 | 6 | 6 | 6 | 5 | 7 |
| *IeeAsesores* | 5 | 5 | 5 | 7 | 7 | 7 | 6 | 7 | 7 | 5 | 5 | 6 | 7 | 7 |

GDPRValidator is quite good. In other words, SMEs value the fact that GDPRValidator can help them to improve GDPR compliance when they process customers' data.

2. Perceived ease of use. In this case, the evaluation obtained is similar to that of the previous variable. Both companies provided values above five. Therefore, we can conclude that the degree to which SMEs expect GDPRValidator to be easy to use is good or very good.

3. Future use. The assessment by the SMEs of the future use of GDPRValidator can be regarded as good. Both companies even rated the use of this tool with a seven (strongly agree) compared with other similar tools to assess their compliance with the GDPR.

## Discussion

This section presents a discussion of the GDPRValidator validation. We answer the research questions presented in "Research questions" and discuss the main threats to validity.

### Research questions

Now we have validated the tool with SMEs, we proceed to answer the research questions initially formulated:

1. How can SMEs check their GDPR compliance when they use cloud services?

One of the companies evaluating `GDPRValidator` was Netberry, which uses cloud services to store customer and employee data. According to the evaluation conducted by this SME, the tool can help the company in the analysis of its GDPR compliance. With this in mind, `GDPRValidator` offers SMEs a report with recommendations on aspects that `GDPRValidator` identified as weaknesses in compliance with GDPR.

2. How can SMEs improve their GDPR compliance when they use cloud services?

According to the evaluation conducted by the SMEs, the tool can help the company in the improvement of its GDPR compliance. For this purpose, `GDPRValidator` offers SMEs a set of documents that will assist them in achieving GDPR compliance by monitoring their compliance. The company will receive three documents intended for customers. These documents allow the company and its customers to know their rights regarding their data. Furthermore, one of these documents ensures that no data can be accessed without the customer's consent. For the internal control of GDPR compliance, `GDPRValidator` generates four documents. These documents include the RPA, the notice of security breaches, the DPIA, and a report with recommendations on aspects that `GDPRValidator` identified as weaknesses in compliance with GDPR and that the company needs to improve.

3. How can SMEs track their customers' data in the cloud?

Both SMEs evaluated `GDPRValidator` as a way to protect the privacy of their cloud data. This utility also implies a greater control of data traceability when using cloud services. For this purpose, `GDPRValidator` suggests, as an additional recommendation, using sticky policies, which are attached to customers' data and travel with the data. These sticky policies define a field called accessHistory that records the third parties who access the data. We use this field to track the data. In addition, the tool also recommends the use of logs in the data controllers to record the accesses that occur, ensuring the tracking of the data accesses, recording who performs the access, the action performed on it, and the purpose of the access.

4. How can SMEs ensure privacy in the cloud by monitoring and auditing data access?

According to the evaluation carried out by both SMEs, `GDPRValidator` can help them in the process of ensuring the privacy of the data handled in the cloud, improving their monitoring and auditing of data access. For this purpose, `GDPRValidator` defines a set of questions related to data privacy in cloud systems. In this way, depending on the SME's responses to these questions, `GDPRValidator` offers a set of recommendations to guarantee data privacy if any weak point in this regard is detected. SMEs can check their GDPR compliance in cloud environments independently of the general requirements established by GDPR since both aspects, general and cloud-related, are independent and optional. Each company decides whether it wants to check compliance with the general GDPR requirements, with those related to working in cloud environments, or both.

In conclusion, based on the validation results, we argue that `GDPRValidator` can be effective in helping a company comply with GDPR. It is also relevant to highlight that we have leveraged the knowledge and experience of the fourth author, who was the main DPO at Santander Private Banking International in Miami city for four years and was responsible for ensuring the privacy of the personal data handled by the entity in line with GDPR. Currently, he is working in the area of cybersecurity at the bank. The third author is also an expert in compliance, assurance, and certification of ICT applications.

### Threats to validity

This section discusses validity aspects according to the perspectives proposed by *Wohlin et al. (2012)*.

#### 1. Construct validity.

The features of the SMEs considered are relevant since they represent the targeted SMEs. Specifically, Netberry Solutions uses cloud services to store its customer data and manages a large amount of data. IeeAsesores handles a smaller amount of data and works mainly on-premise, but it is still a representative example of many SMEs that need to be GDPR compliant. In addition, for the general GDPR requirements assessment, we have used the checklist provided by the GDPR.EU. It is an official checklist offered by the EU.

#### 2. Internal validity.

We have used a well-known, standard method for technology validation, namely the TAM model. This contributes to internal validity. The SMEs filled out the TAM questionnaire after completing the whole `GDPRValidator` process, which also contributes to obtaining more reliable results (perceptions) than, for example, from companies that have only watched a video. There is a threat in the fact that the companies used the tools on their own, without any control or supervision. Nonetheless, they could ask questions in case of doubts about how to use the tool and its results.

#### 3. Conclusion validity.

We must point out that the conclusions drawn from the validation of `GDPRValidator` could be stronger, as the first scope of validation has been two SMEs. However, we focus on SMEs that treat their data in a way that encompasses the main general aspects of SMEs. In addition, we have paid attention to general results when drawing conclusions, such as the overall results for perceived usefulness, not to specific, detailed answers.

#### 4. External validity.

As stated, it is important to highlight that the characteristics of the companies in the evaluation are relevant and representative of SMEs. We have considered a larger company, Netberry Solutions, which uses cloud services to store its customers' data, as well as IeeAsesores, which is a smaller company that works mainly on-premise. Since these companies handle customer and employee data similarly to most SMEs, they can serve as representative of many SMEs. Nonetheless, it would be necessary to collect feedback from further SMEs to ascertain whether any variation in the results could occur.

## CONCLUSIONS

In this article, we have presented the `GDPRValidator` tool, which is designed to help SMEs be GDPR compliant. Many SMEs take advantage of the cloud to offer their services and want to expand to international markets. As a result, they must protect their customers' data to prevent it from being compromised, which would lead to their customers losing trust in the company and the possibility of substantial fines.

Most SMEs are unaware of what constitutes personal information and whether it is stored on the local machine or in the cloud. In addition, they lack the resources and technical expertise to locate and identify such data. GDPR, the EU regulation for data protection, establishes the legal framework for protecting the personal data of European citizens. It defines guidelines to help companies to protect the data they manage. In this work, we have proposed a tool to help non-legally trained SMEs in the process of guaranteeing the privacy of their customers' data. For this purpose, `GDPRValidator` provides them with several documents. Furthermore, this tool is aimed at SMEs since their budget is usually insufficient to allow the hiring of legal experts. However, it could also be beneficial for DPOs working for bigger companies as it could facilitate their work.

In the implementation of `GDPRValidator`, we have considered the main concepts of GDPR, such as the purpose of data access, consent to access, DPIA, or data management audits, taking into account general and cloud-related GDPR requirements. In addition, some additional recommendations are given to facilitate SMEs' GDPR compliance process.

The `GDPRValidator` tool generates a set of documents and several templates necessary to manage the GDPR compliance process. Some of these documents are templates for *DPIA* or requests for consent for third parties to access data containing the SME's details.

Thus, an SME can more easily understand the documentation that it must handle in complying with GDPR, which facilitates the steps to follow without knowing the regulation or consulting an expert. However, although our tool provides general information, in the case of having specific doubts about certain special situations, the help of a professional may be necessary to resolve them, even though consultations with experts usually come at a high price. In any case, by using `GDPRValidator`, companies would have the documentation needed to be GDPR compliant and reduce the cost of these legal consultations.

In the future, we plan to extend our tool to perform a complete data risk analysis. We also intend to add the functionality of estimating a GDPR compliance level, so that SMEs that implement the `GDPRValidator` recommendations will be able to see how they have improved their compliance. A method to perform this calculation objectively will need to be explored, and this will involve consulting various regulatory experts on how best to do so.

## APPENDIX A

All questions of the checklist for GDPR compliance of the general GDPR compliance guidelines:

**1. Lawful basis and transparency.** According to article 6 and recital 40, personal data should only be processed with the consent of the data subject, in this case the customers, or

on another legitimate basis. Recital 58 and article 12 regulate the transparency of information. Some related questions on the checklist are the following:

**(a)** *Does the company conduct an information audit to determine what information is processed and who has access to it?*

**Report suggestion:** The most effective way to demonstrate GDPR compliance is to conduct a data protection impact assessment (DPIA), article 35. Among other things, the DPIA must include the purpose of the processing, the types of data you process, who has access to it in the organization, any third parties (and their location) who have access, and how you plan to protect the data (such as encryption), and when you plan to erase it (if possible). The CSP must provide your company with all this data access information. Therefore, the CSP should prepare its DPIA to keep track of all cloud accesses. This issue can be agreed upon when signing the SLA with the CSP. Your company will need to ask the CSP for their DPIA to complete their DPIA.

The `GDPRValidator` framework allows companies to assess whether they need a Data Protection Impact Assessment *GDPR.EU (2022b)* (DPIA), see "Data Protection Impact Assessment", and generates a draft of it.

**(b)** *Is there a legal justification for your data processing activities?*

**Report suggestion:** The possible answers to this question relate to the legal basis for the processing defined by article 6. When the legal basis is "consent", that is, the data subject (in your case, your customer) has given consent to data access, your company has additional obligations that include offering data subjects the opportunity to revoke this consent. In the case of processing for "legitimate interests" pursued by the others (your company, the CSP, or by a third party), article 6. f, an assessment of privacy impacts must be conducted by your company. There are other provisions in articles 7 to 11 related to children and special categories of personal data.

**(c)** *Does the company provide clear information about your data processing and legal justification in your privacy policy?*

**Report suggestion:**

This information should be included in your privacy policy and provided to data subjects when their data are collected. Data subjects must be informed about the data collected about them and why, according to article 12. You must report how the data is processed, who has access to it, and how it is kept safe.

**2. Data security.** Whenever personal data is processed, the company needs to follow the data protection principles outlined in article 5. Some of the questions that allow us to evaluate the security of data processing are the following:

**(a)** *From the moment projects begin to develop products to each time data are processed, does the company consider data protection at all times?*

**Report suggestion:** The company should evaluate whether the company applies the principles of "data protection by design and default" defined in article 25, such as implementing "appropriate technical and organizational measures".

**(b)** *When a data breach occurs, does your company have a procedure for contacting the authorities and your data subjects?* It is important to consider that, according to GDPR, a data breach that exposes personal data must be reported to the supervisory authority within 72 h.

**Report suggestion:** Here, we provide a list of the EU member states' supervisory authorities. And we include in the recommendation report the following: According to article 34, you must also notify the supervisory authority and your customers immediately, within 72 h, if there is a data breach. Unless the breach is unlikely to put them at risk, for instance, if the data stolen is encrypted.

**(c)** *Whenever possible, does the company protect personal information by encryption, pseudonymization, or anonymization?*.

**Report suggestion:** Your company must implement cryptography or pseudonymization as often as possible under the GDPR.

**(d)** *Does your company create an internal security policy for your team members and build awareness about data protection?*

**Report suggestion:** The GDPR requires additional training for employees with access to personal data and non-technical employees. The provision of training courses for your company's staff is recommended.

**(e)** *Has your company established a process for conducting a data protection impact assessment?* We do not need to consider this question, because it is related to question 1.a.

**3. Accountability and governance.** Accountability for data protection refers to taking responsibility for actions and decisions regarding data protection, while governance refers to controlling and directing data protection. The two concepts are interdependent, article 5.

**(a)** *Has your company signed a data processing agreement with any third parties who process personal data on your behalf?*

**Report suggestion:** For GDPR compliance, almost all services require a standard data processing agreement, which includes any third-party services that handle your data subjects' personal information. A template for this agreement is available at https://gdpr. eu/data-processing-agreement/, and it must be visible on your website and specify the rights and obligations of each party. It is imperative to only work with third parties who can provide adequate data protection guarantees. Thus, the SLA with the CSP must specify the list of permitted third parties.

**(b)** *Has your company appointed a Data Protection Officer (if necessary)?*

**Report suggestion:** There are some circumstances in which organizations are required to have a Data Protection Officer (DPO). These conditions usually arise when a company's activity involves large-scale data processing or sensitive or special categories of data are processed, according to articles 9 and 10. A DPO should be an expert in data protection. This officer should have duties that include monitoring GDPR compliance, assessing data

protection risks, advising on data protection impact assessments, and cooperating with regulators. In any case, although having a DPO is not mandatory, it is recommended.

**(c)** *Has your company appointed someone to ensure GDPR compliance?*

**Report suggestion:** According to the guidelines of "data protection by design and by default", article 25, GDPR compliance must be assigned to someone within the organization. Data protection policies must be evaluated and implemented by this individual.

**(d)** *Does your company appoint a representative within an EU member state if it is not an EU member?*

**Report suggestion:** If you process data relating to individuals in a particular country, you must designate a representative there who can communicate with the data protection authority on your behalf, article 27. In cases where EU citizens are affected by processing in the multiple Member States, the GDPR does not provide guidance. In this case, it may be prudent to designate a representative in a member state that speaks your language.

**4. Privacy rights.** Following article 12, the controller shall facilitate the exercise of data subjects' rights under articles 15 to 22. A controller may not deny the customers' rights unless the controller can demonstrate that it is not in a position to identify them, article 11.

**(a)** *Can your customers easily access all the information you have about them?*

**Report suggestion:** First, your organization should ensure that the client requesting the data is verified. Your customers have the right to know what personal information your company has about them, how it is used, how long it is stored, and why it is stored, article 15. You must respond to these requests within one month, article 16. This information is sent for free the first time it is requested, but subsequent copies may have a cost.

**(b)** *Do your customers have the ability to update inaccurate or incomplete information easily?*

**Report suggestion:** First, your company needs to check the identity of the customer asking for the information. Make sure your company has a process that allows customers to audit their data and update their personal information if necessary, article 15. Your company must respond to these requests within one month, article 16.

**(c)** *Do your customers have the option of deleting their personal information?*

**Report suggestion:** It is also necessary to verify the identity of the person making the request. Customers have the right to request the deletion of all their data held by your company, article 17, also called the right to be forgotten. You can refuse the request in certain circumstances, such as exercising freedom of speech or fulfilling a legal obligation. These requests must be answered within one month, article 16.

**(d)** *Can your customers ask you to stop processing their data easily?*

**Report suggestion:** Several grounds allow your customers to restrict or stop the processing of their data, see article 18. Usually, this happens when there is a dispute over whether the processing is lawful or accurate. However, you can still store their data while processing is

restricted. The customers must be notified before you resume processing their data. Your company has to send this notification within a month.

**(e)** *Do your customers have the option of receiving a copy of their data in a form that can be easily transferred to a different company?*

**Report suggestion:** Therefore, your organization should be able to send the customer data in a commonly readable format, either directly to them or to a third party they choose, article 20. Your customer data may be given to your competitor, which may seem unfair from a business perspective. However, from a privacy perspective, the idea is that people own their data, not you.

**(f)** *Do your customers have the option to object to the processing of their data?*

**Report suggestion:** If your organization processes customers' data for direct marketing, it must cease doing so immediately for that purpose, article 21. On the other hand, you may be able to challenge their objection if you demonstrate "compelling legitimate grounds".

**(g)** *Does your organization have procedures to protect individuals whose rights are affected by automated processes?*

**Reporting suggestion:** Some organizations use automated processes for decisions that affect people with legal effects or "of similar importance", article 22. If you think that applies to your organization, you need to create procedures to guarantee their rights and protect their freedoms and legitimate interests. Your company must make it easy for customers to request human intervention, evaluate decisions, and question them.

**5. Third Parties.**

**(a)** *Do you have third parties requesting "consent" to access your customers' data that are not included as recipients in your contracts?* For each of these companies, please enter the following information: company category, the purpose for access, action, duration of storage, in the case the data is copied.

**Report suggestion:** Your company must have the identification of those companies that wish to access your customer's data before the "consent" is given, as well as the purpose of the access, the action to be carried out on the data and, in the case of copying this data, the duration of storage of that copy.

All questions in the checklist for GDPR compliance of the cloud-related GDPR compliance guidelines:

**1. Contracts.**

**(a)** *Has your company signed an SLA with the CSP to be responsible for the treatment of personal data on your behalf?*

**Report suggestion:** Your company must sign an SLA with the CSP responsible for processing your customers' data, acting as the joint controller, article 26. The SLA should identify the responsibilities of the SME and the CSP, as controller and joint controller, in a transparent manner. Specifically, concerning the exercise of rights of the interested party (customers) and their respective responsibilities to provide the information referred to in

articles 13 and 14, the agreement may designate a point of contact for these interested parties. It is important to keep contracts updated.

Furthermore, your company needs to ensure that the hired CSP must declare adherence to the EU Cloud Code of Conduct (EU Cloud COC) *Calder (2021)* about its cloud service, article 40. It provides cloud-specific approaches and recommendations, including a road map that tracks code requirements to GDPR and international standards such as ISO 27001 *Beckers (2015)* and 27018 *de Hert, Papakonstantinou & Kamara (2016)*[5]. In addition, the CSP and the competent supervisory authority must certify that the cloud service fully complies with the provisions in the code.

**(b)** *Has the CSP signed an SLA for data processing with any third parties who access/ process personal data?*

**Report suggestion:** For GDPR compliance, almost all services require a standard data processing agreement, which includes any third party services that handle your data subjects' personal information. It is imperative to only work with third parties who can provide adequate data protection guarantees. This suggestion complements the one in the point 3 of the checklist related to *Accountability and governance* in the general GDPR compliance guidelines.

**2. Duration.**

**(a)** *Does your company establish a date for data retention?*

**Report suggestion:** The storage of data should not exceed the necessary storage time. The SLA signed with the CSP must specify the processing duration. After the retention period has expired, all data, including copies, must be deleted.

**3. Information.**

**(a)** *Does your company specify a procedure to notify when a breach is detected?*

**Report suggestion:** GDPR in the article 4 determines what is considered a breach in the collection, retention, and processing of data: "the accidental or unlawful destruction, loss, alteration, unauthorized disclosure of, or access to, personal data transmitted, stored or otherwise processed". Then, the company must specify the procedure that it follows to notify when a breach is detected, for instance, in the event of theft or a leak. Before a breach is reported in the media, the CSP should inform your company, and your company should notify its customers and supervisory authorities. According to article 36, the controller must report any breach of personal data to the supervisory authority competent under article 55 within 72 h of becoming aware of it, unless the breach is unlikely to result in a risk to the rights and freedoms of natural persons. This suggestion complements the related one in point 2a in *Data security* in the general GDPR compliance guidelines.

**(b)** *Does the CSP explicitly inform your company where each customer's data is stored?*

**Report suggestion:** All data of European citizens should either be stored in the EU, so that they are subject to European privacy law, or in a place where the same level of protection is

[5] The International Organization for Standardization and International Electrotechnical Commission (ISO/IEC) sets out the requirements for an information security management system (ISMS) by the definition of the standard ISO/IEC 27001. To ensure the protection of "personally identifiable information" processed by cloud service providers, ISO/EIC published ISO/IEC 27018 (ISO 27018) in 2014.

guaranteed. Your company is responsible for arranging with the CSP the location for storing the customers' data. The SLA, signed between the CSP and the company, should include this information.

**4. Privacy rights.** Some of these questions are related to the ones shown in the general GDPR compliance guidelines (Privacy rights 6) but applied in the cloud context.

**(a)** *Can your company access and delete the customers' data from the cloud?*

**Report suggestion:** Your company has the right to access its customers' data and have them deleted, the right to access (article 15), and the right to erasure ("to be forgotten", article 17). The CSP must facilitate this process by making the data available to your company and your customers in a structured format. It is also important to consider data backups when deleting customer data.

**(b)** *Does your company retain ownership and control of your customers' data at all times?*

**Report suggestion:** As controller, your company must retain ownership and control of your customers' data at all times. To ensure that your company retains ownership and control of your customers' data at all times, the corresponding SLA should include this condition.

**(c)** *Does your company have information about the metadata that the CSP collects?*

**Reporting suggestion:** Your company should ask the CSP what metadata it collects and whether your company has a right to opt out. The CSP must guarantee that it will inform your company about this metadata, so this condition should also be included in the contract.

**5. Data Protection Impact Assessment (DPIA).**

**(a)** *Does the CSP have a Protection Impact Assessment (DPIA) that determines the risks associated with cloud hosting or services?*

**Report suggestion:** The CSP, as a joint data controller, should also have a DPIA (see "Data Protection Impact Assessment") that determines the risks associated with cloud hosting or services. It is recommendable, but not mandatory. The CSP must send this DPIA to your company every time that it is updated. This suggestion complements the related one in point 1a, under *Lawful basis and transparency* in the general GDPR compliance guidelines.

**6. Third Parties.**

**(a)** *Do you have third parties in the cloud requesting "consent" to access your customers' data that are not included as recipients in your contracts?* For each of these companies, please enter the following information: company category, the purpose for access, action, and duration of storage, in the case of the data being copied.

**Report suggestion:** Your company must have the identification of those companies that wish to access your customer's data before the "consent" is given, as well as the purpose of

the access, the action to be carried out on the data and in case of copying this data, the duration of storage of that copy.

### Funding
This work was supported by the following Spanish Ministry of Science and Innovation (co-financed by European Union FEDER funds) projects: "FAME (Formal modelling and advanced testing methods. Applications to medicine and computing systems)", reference RTI2018-093608-B-C32, and "AwESOMe project (Advanced methodologies for architectures, designing and testing of software systems)", reference PID2021-122215NB-C32. There was also support from the Junta de Comunidades de Castilla-La Mancha project SBPLY/17/180501/000276/01 (co-funded with FEDER funds, EU) and from the Ramon y Cajal Program (Grant RYC-2017-22836 funded by MCIN/AEI/ 10.13039/ 501100011033 and by "ESF Investing in your future"). The funders had no role in study design, data collection and analysis, decision to publish, or preparation of the manuscript.

### Grant Disclosures
The following grant information was disclosed by the authors:
Spanish Ministry of Science and Innovation (co-financed by European Union FEDER funds) Projects: RTI2018-093608-B-C32.
AwESOMe Project: PID2021-122215NB-C32.
Junta de Comunidades de Castilla-La Mancha Project: SBPLY/17/180501/000276/01.
Ramon y Cajal Program: RYC-2017-22836, MCIN/AEI/ 10.13039/501100011033.

### Competing Interests
M. Emilia Cambronero is an Academic Editor for PeerJ. David Cebrián is employed by Santander Private Banking International.

### Author Contributions
- M. Emilia Cambronero conceived and designed the experiments, performed the experiments, analyzed the data, performed the computation work, prepared figures and/or tables, authored or reviewed drafts of the article, and approved the final draft.
- Miguel A. Martínez performed the experiments, analyzed the data, performed the computation work, prepared figures and/or tables, authored or reviewed drafts of the article, and approved the final draft.
- José Luis de la Vara analyzed the data, authored or reviewed drafts of the article, and approved the final draft.
- David Cebrián analyzed the data, performed the computation work, authored or reviewed drafts of the article, and approved the final draft.
- Valentín Valero analyzed the data, authored or reviewed drafts of the article, and approved the final draft.

## Data Availability

The code is available at Zenodo: MiguelMartinez47. (2022). MiguelMartinez47/GDPRValidator: GDPRValidator v1.0 (v1.0). Zenodo. https://doi.org/10.5281/zenodo.7224749.

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
