# Peer review of "GDPRValidator: a tool to enable companies using cloud services to be GDPR compliant"

_PeerJ Computer Science, doi:10.7717/peerj-cs.1171_

## Round 0.1 · original submission · Major Revisions

The paper needs further improvements. The authors should address the comments from the reviewers in the revised paper.

Reviewer 1 ·

Basic reporting

Software Compliance with the law and regulations can be a difficult undertaking, especially for multinational software firms with a clientele from a variety of industry sectors and locations. First, even though it is on a very tiny scale, there might be a significant number of regulatory compliance requirements that must be taken into account. Given the tremendous breadth of federal regulation, it is not possible to list the complete extent of U.S. law dealing to data, privacy, and records. This paper introduces a tool called GDPRValidator that can help cloud service providers manage and store employee or customer data in the cloud while remaining GDPR compliant. This tool is intended for small and medium-sized businesses (SMEs) that have moved all or a portion of their services to the cloud in order to benefit from this technology and gain a competitive edge. The following comments are suggested for improvements of the content.

Experimental design

1- Abstract: The abstract is too general and need to specify the main results and findings of the proposed tool.

2- Motivation: What is the motivation of this work? Motivation behind using the tool, is not clear.

3- Language: The language usage throughout this paper need to be improved, the author should do some proofreading on it. Also, the format of the whole paper looks messy, the authors should put the format into a unified form. Give the article a mild language revision to get rid of few complex sentences that hinder readability, and eradicate typo errors.

4- Literature review: It is not enough discussed in this article, highly appreciate to reinforce this part. The related work could be extended and incorporates more comprehensive discussions on topics in the software compliance with different regulations. I suggest summarizing the related work in a form of table.

5- Difference: The difference of the proposed tool with existing tools (if any) is not properly addressed. These differences should be highlighted.

Validity of the findings

6- Theoretical analysis: What is the efficiency of the tool? Do the authors address this? any theoretical analysis or experimental analysis? is it real-time?

Additional comments

Software Compliance with the law and regulations can be a difficult undertaking, especially for multinational software firms with a clientele from a variety of industry sectors and locations. First, even though it is on a very tiny scale, there might be a significant number of regulatory compliance requirements that must be taken into account. Given the tremendous breadth of federal regulation, it is not possible to list the complete extent of U.S. law dealing to data, privacy, and records. This paper introduces a tool called GDPRValidator that can help cloud service providers manage and store employee or customer data in the cloud while remaining GDPR compliant. This tool is intended for small and medium-sized businesses (SMEs) that have moved all or a portion of their services to the cloud in order to benefit from this technology and gain a competitive edge. The following comments are suggested for improvements of the content.

1- Abstract: The abstract is too general and need to specify the main results and findings of the proposed tool.

2- Motivation: What is the motivation of this work? Motivation behind using the tool, is not clear.

3- Language: The language usage throughout this paper needs to be improved, the author should do some proofreading on it. Also, the format of the whole paper looks messy, the authors should put the format into a unified form. Give the article a mild language revision to get rid of few complex sentences that hinder readability and eradicate typo errors.

4- Literature review: It is not enough discussed in this article, highly appreciate to reinforce this part. The related work could be extended and incorporates more comprehensive discussions on topics in the software compliance with different regulations. I suggest summarizing the related work in a form of table.

5- Difference: The difference of the proposed tool with existing tools (if any) is not properly addressed. These differences should be highlighted.

• Theoretical analysis: What is the efficiency of the tool? Do the authors address this? any theoretical analysis or experimental analysis? is it real-time?

Reviewer 2 ·

Basic reporting

The paper is written in detailed manner. The authors have made a vast study in the subject area and the information is clear for readers to understand the need for the current research.

Experimental design

The fundamental aim of the validation is to study whether GDPRValidator is a feasible means of enabling
SMEs that use cloud services to assess GDPR compliance. The subsection 7.2 discussed results. The results seems to be fine.

Validity of the findings

The fundamental aim of the validation is to study whether GDPRValidator is a feasible means of enabling
SMEs that use cloud services to assess GDPR compliance. The subsection 7.2 discussed results. The results seems to be fine.

Additional comments

Overall the paper is fine to be accepted.

·

Basic reporting

The main contribution is the presentation of a tool to help SMEs understand and validate
their compliance with GDPR. The paper is written well and in a elaborated way. This puts forth that the authors have worked strongly in this area and this is reflected by the implementation details provided as a link for reference. Overall the work reported is fine.

Experimental design

The Implementation is done by considering the purpose of data access, consent to access, DPIA, or data management audits, taking into account general and cloud-related GDPR requirements.

The application reported at http://pluton.i3a.uclm.es:8080/companyDataForm2.html is fine and good. It reflects a quality contribution.

Validity of the findings

A tool is proposed to help non-legally trained SMEs in the process of guaranteeing the privacy of their customers’ data. To achieve privacy, GDPRValidator provides them with several documents. Furthermore, this tool is aimed at SMEs since their budget is usually insufficient to allow the hiring of legal experts.

Additional comments

The reported contribution GDPRValidator may be accepted.

---

## Round 0.2 · Minor Revisions

Comments from the Academic Editor:

The paper still needs further improvements to increase the paper quality and so the following notes should addressed:

1. The authors should include a paper road map in Introduction Section.
2. They should look at the recent report from General Data Protection Regulation (GDPR) and cited in the paper.
3.The following research question:
How can SMEs check and improve their GDPR compliance when using cloud services?
Has two answers and so I suggest to divide this research question to questions.
If not, please clarify why.
4. I think Contribution Subsection can be rephrased to one or two paragraphs.
5. The authors should avoid the repeated paragraphs and statements mentioned in Abstract, Introduction and Related work Sections.
6. What are the main advantages of the proposed tool over other tools such as FACILITA 2.0, Facilita – Emprende and LogicGate? These should be mentioned clearly in the related work.
7. Is the methodology of the proposed tool a repeated cycle? I noticed the arrow at the bottom of the methodology figure towards from the end to step no 7. If not, the authors should clarify this methodology more.

Reviewer 1 ·

Basic reporting

The authors have satisfied all the comments of previous round successfully.

Experimental design

The authors have satisfied all the comments of previous round successfully.

Validity of the findings

None

Additional comments

The authors have satisfied all the comments of previous round successfully. I recommend accepting this paper.

---

## Round 0.3 · accepted · Accept

I'm happy to accept the paper now without further revisions - Well Done.